# Dissociative Electron Attachment Cross Sections for Ni(CO)$_4$, Co(CO)$_3$NO, Cr(CO)$_6$

**Maria Pintea** [1,*] , **Nigel Mason** [1,*] **and Maria Tudorovskaya** [2]

[1] School of Physical Sciences, University of Kent, Canterbury CT2 7NZ, UK
[2] Quantemol Ltd., London EC1V 2NZ, UK
[*] Correspondence: mp675@kent.ac.uk (M.P.); n.j.mason@kent.ac.uk (N.M.)

**Abstract:** Ni(CO)$_4$, Cr(CO)$_6$, Co(CO)$_3$NO are some of the most common precursors used for focused electron beam induced deposition. Some of the compounds, even though extensively used have high requirements when it comes to handling being, explosives, highly flammable and with high toxicity levels, as is the case of Ni(CO)$_4$. We are employing simulations to determine values hard to determine experimentally, and compare them with DFT calculations and experimental data where available. The use of Quantemol-N cross section simulations for dissociative electron attachment (DEA) at low electron energy in the range of 0–20 eV, gives valuable information on the fragmentation of the molecules, based on their bond dissociation energies, electron affinities and incident electron energies. The values obtained for the cross sections are $0.12 \times 10^{-18}$ cm$^2$ for Ni(CO)$_4$, $4.5 \times 10^{-16}$ cm$^2$ for Co(CO)$_3$NO DEA cross-sections and $4.3 \times 10^{-15}$ cm$^2$ for Cr(CO)$_6$.

**Keywords:** DEA; cross sections; FEBID; dissociative ionization; excitation cross-sections





## 1. Introduction

As Focused Electron Beam Induced Deposition (FEBID) [1–3] is developing with the possibility of becoming a viable manufacturing technique, the need to have more data available on molecules of interest, alkenes, silanes, metal halogens, carbonyls, phosphines, acetylacetonates [1], increases. The limitation of this direct-write fabrication technique comes in the appearance of secondary and backscattered electrons as part of the primary electron beam and secondary electron beam. The effect of the secondary and backscattered electrons at low electron energy level, 0–50 eV, is the deposition of a thin halo and creation of secondary structures in the vicinity of the primary structures as well as incomplete fragmentation of the precursors reducing the purity of the final structure [2]. To analyze and recreate these effects, gas phase and surface science studies are employed. In gas phase studies, the interaction of molecules with single electrons is evaluated and resulting fragmentation pathways analyzed. On the other hand, in the ultra-high vacuum surface science setups, the interaction of the electrons with the molecules and the substrates is evaluated, identifying the species desorbing from the substrate. However, for many of these FEBID compounds an increase in information on the probability of collision between electrons and molecules and the dynamics of the processes underlying the induced chemistry in organometallic compounds, is needed. Thus the cross sections for dissociative ionization, elastic scattering, vibrational excitation, dissociative electron attachment, neutral dissociation and bipolar dissociation are required. The data presented from our R-matrix calculations will focus on three widely used compounds in Focused Electron Beam Induced Deposition (FEBID), Ni(CO)$_4$ (Figure 1), Co(CO)$_3$NO and Cr(CO)$_6$.

*Molecular Complexes Used for Calculations*

The carbonyl group compounds (Ni(CO)$_4$, Cr(CO)$_6$ and Co(CO)$_3$NO) have simple symmetric structures that make them suitable for electron-induced chemistry applications and potentially suitable for creating very clean deposits in the focused electron beam

induced process with purities over 90% and low resistivity. The structural representation of the compounds is presented in Figure 2 and the X, Y, Z coordinates used for Quantemol-N calculations are presented in the additional information in Appendix A.

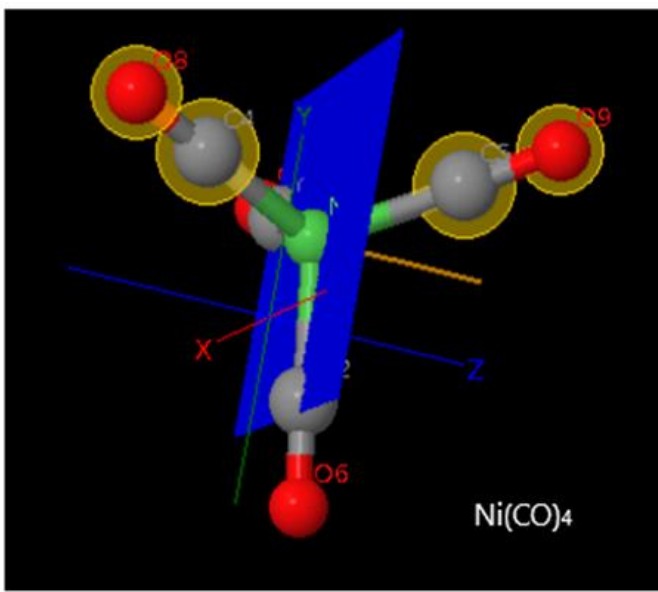

**Figure 1.** $Ni(CO)_4$ structural symmetry.

**Figure 2.** Molecular complexes, $Ni(CO)_4$, $Co(CO)_3NO$ and $Cr(CO)_6$.

$Ni(CO)_4$ is less used in the industry due to its high toxicity, similarly to $Cr(CO)_6$. It is highly flammable and insoluble in water. Its lowest decomposition temperature is 50 °C, making it hard to be used at room temperature and requiring cryogenic conditions. The $Cr(CO)_6$ compared to the $Ni(CO)_4$ has higher volatility and is more stable at room temperature with a thermal decomposition point over 150 °C. Similar to the $Cr(CO)_6$, the $Co(CO)_3NO$ has a high decomposition temperature, around 130–140 °C. Detailed Raman, FTIR and MOs studies are compared in the Cross Sections section with the theoretical model obtained from our calculations.

## 2. R-Matrix Theory and Method of Calculation

Scattering cross sections calculations based on the R-matrix theory [4] were carried out to determine the cross-sections of electron scattering [5,6] by $Ni(CO)_4$, $Cr(CO)_6$ and $Co(CO)_3NO$. We used the Quantemol-N simulation package interfacing UKRmol code suite [7]. Quantemol-N can be used to compute a number of cross-sections for elastic and inelastic electron scattering, the most important in the electron beam induced chemistry being the dissociative electron attachment (DEA) cross-sections [8].

The DEA process is a process widely found in nature resulting in the formation of negative ions at relatively low electron energies. The dissociation process and the induced chemical reactions follow the schematics:

$$AB + e^- \rightarrow \mathbf{AB^*} \rightarrow AB^- \rightarrow \mathbf{A^-} + B \text{ or } A + \mathbf{B^-} \tag{1}$$

which can be followed by the dissociation reaction:

$$M(CO)_n \rightarrow M(CO)_{n-m} + m(CO), \text{ where } M = Cr, Ni, Co \tag{2}$$

The DEA process can undertake two paths in the evolution of the compound from gas—phase molecules to fragments or radicals: the first path (a) is the irradiation of the molecule with an electron that would cause a change in the molecule's energy found in a certain resonant state, transitioning to a higher excited state, and the second path (b) is either a repeated autoionization or if the resonance state has a longer lifetime, dissociation of the molecule into fragments.

The total dissociative attachment cross section is a weighted sum of DEA cross-sections, $\sigma_i$, over all resonances in the scattering process (3):

$$\sigma_T(E) = C \sum_i S_i \ \sigma_i(E) \tag{3}$$

The positions of the resonances found with the UKRmol routine RESON [9]. S is the survival probability for the resonance or the probability of a resonance, E is the incident electron energy, and C is the adjustment factor [8]. It is important to note that in the model used by Quantemol-N, the fragments are moving in the effective potential and therefore behave like a quasi-diatomic molecule. For each resonance, the partial cross-section has the Breit–Wigner shape:

$$\sigma_{BW}(E, r) = (2\pi/E) \ (\Gamma^2/4)/[(E - V(r))^2 \ (\Gamma^2/4)] \tag{4}$$

where $\Gamma$ is the width of the resonance, r is the distance between the dissociating fragments, and V(r) is the effective potential.

The resonance positions and widths are determined by the UKRmol codes which treat the incoming electron in the same way as the molecule's electrons in its vicinity inside the so-called R-matrix sphere. The scattering wavefunction can be expanded over the target states and expressed in terms of N-electron target wavefunctions, continuum orbitals representing the scattering electron inside the R-matrix sphere, and additional quadratic integrable functions constructed from the target occupied and virtual molecular orbitals [4]. Far away from the molecule, the scattered electron moves in the effective potential.

Whilst the ab initio part of the calculation is quite rigorous, further assumptions about the dissociation channels and assigning resonances to a specific channel may introduce uncertainty.

The cross section values from Quantemol-N software are often in very good agreement with the experimental data available for the particular molecules. In preparing the present calculations we benchmarked the current calculations producing a set of $CH_4$ cross section data, we obtained similar results for the total cross sections and inelastic cross sections, which in previous Quantemol-N calculations replicated experimental data.

### 3. Computational Details

The present work was performed using Quantemol-N for cross-section determination with a separate set of parameters for each of the molecules. The $Cr(CO)_6$ was run using CAS 2 at 10 eV cut-off and R-matrix radius of 13 with the work point group symmetry in $D_{2h}$ and user defined basis set based on 6-31* for O and C and cc-pVTZ for Cr.

The $Ni(CO)_4$ molecule was defined with the initial parameters CAS 3, 10 eV cut-off and an R-matrix radius of 13 with a symmetry work point group of C2h and a user defined basis set, 6-31* for O and C and cc-pVTZ for Ni.

For the $Co(CO)_3NO$ parameters such as CAS 2, 10 eV cut-off, a R-matrix radius of 12 with the molecule in Cs symmetry work point group and user defined basis set 6-31* for O and C and cc-pVTZ for Co.

## 4. Cross Sections

**$Co(CO)_3NO$.** The $Co(CO)_3NO$ compound has $C_{3v}$ symmetry, with the three CO groups on the faces of a tetrahedron and the NO group to Co(IV). The total cross section is high with values of $1.4 \times 10^{-12}$ cm$^2$ for an energy range of 0–100 eV. Engmann et al. (2013) [10] measured the DEA cross section obtaining a maximum value of $4.1 \times 10^{-16}$ cm$^2$ for the loss of only one CO ligand, this being the predominant DEA process, while the measured DI cross section has a value of $4.6 \times 10^{-16}$ cm$^2$.

The spectroscopy of $Co(CO)_3N^{15}O$ and $Co(CO)_3N^{14}O$ in vapor form has been analyzed [11], with the infrared spectrum showing vibrational band frequencies at 2108 cm$^{-1}$, 2047 cm$^{-1}$, 1822 cm$^{-1}$ for C-O and N-O stretch and 2010 cm$^{-1}$ for $C^{13}$-O isotopic species of $Co(CO)_2(C^{13}O)NO$. Bartz et al. (1998) [12] determined the highest excited state of $Co(CO)_3NO$ y2F5/2 at 36300 cm$^{-1}$ equivalent to 103.8 kcal/mol, the three Co-CO bonds and one Co-NO bond needing an extra energy of 154.4 kcal/mol (Table 1) to break the ligands and 38 kcal/mol as the adiabatic metal-ligand bond dissociation energy for two CO groups.

**Table 1.** Bond dissociation energies for $Co(CO)_3NO$, $Ni(CO)_4$, $W(CO)_6$.

| Compound | Dissociation | BDE (kcal/mol) |
|---|---|---|
| $Co(CO)_3NO$ | $Co + 3CO + NO$ | 144.8–154.4 [12] |
| $Ni(CO)_4$ | $Ni(CO)_3 + CO$ | 35 [13], 22.3 [14] |
| $Co(CO)_3NO$ | $Co(CO) + 2CO + NO$ | 115 [12] |
| $Cr(CO)_6$ | $Cr(CO)_5 + CO$ | 49.8 [15], 38 [16] |

At higher energy, the study of Rosenberg et al. (2013) [11] splits the problem in two parts, for irradiation with electron densities less than $5 \times 10^{16}$ e$^-$/cm$^2$ where the C(1s) peaks appear at 287.8 eV and 293.3 eV, a π-π* transition, the N(1s) that has the peak at 401.6 eV, the O(1s) peaks at 534.0 eV and 534.6 eV and an oxide peak at 529.7 eV, and the $Co(2p_{3/2})$ peak at 780.9 eV. For electron densities over $5 \times 10^{16}$ e$^-$/cm$^2$, the C(1s) peak appears only at 285 eV, the N(1s) peak does not change its position, the O(1s) peaks attenuate in amplitude and the oxide peak at 529.7 eV now increases in amplitude. If the concentration ratios of after and before irradiation are taken into account, a decrease in the values of $O/O_0$ and $C/C_0$ is observed and $N/N_0$ remains constant.

The DEA process is presented in showing the sequential nature of the fragmentation (5):

$$
\begin{aligned}
Co(CO)_3NO \rightarrow Co(CO)_3NO^* \rightarrow Co(CO)_3NO^- &\rightarrow Co(CO)_2NO^- + CO \rightarrow Co(CO)NO^- + 2(CO) \\
&\rightarrow Co(CO)_3{}^- + NO \rightarrow Co(CO)_2{}^- + NO + CO \\
&\rightarrow Co(CO)^- + NO + 2CO \\
&\rightarrow Co^- + NO + 3(CO)
\end{aligned}
\tag{5}
$$

Knowing that the dissociative electron attachment process happens in the range of 0–15 eV (see Table 2), negative ions are formed at incident electron energies between 0.2 eV to 7.1 eV, all relatively low electron energies with values under 10 eV. The formation of pure Co$^-$ takes place at 7.1 eV, stripping off a $(CO)_3NO$ radical. The dissociative electron attachment to $Co(CO)_3NO$ for the dissociation of one CO ligand at 0.9 eV with the formation of $Co(CO)_2NO^-$ ion is presented in Figure 3 with values of ~22.5 Å. The incident bond dissociation energy for the formation of all negative ions used in our calculations is presented in Table 2.

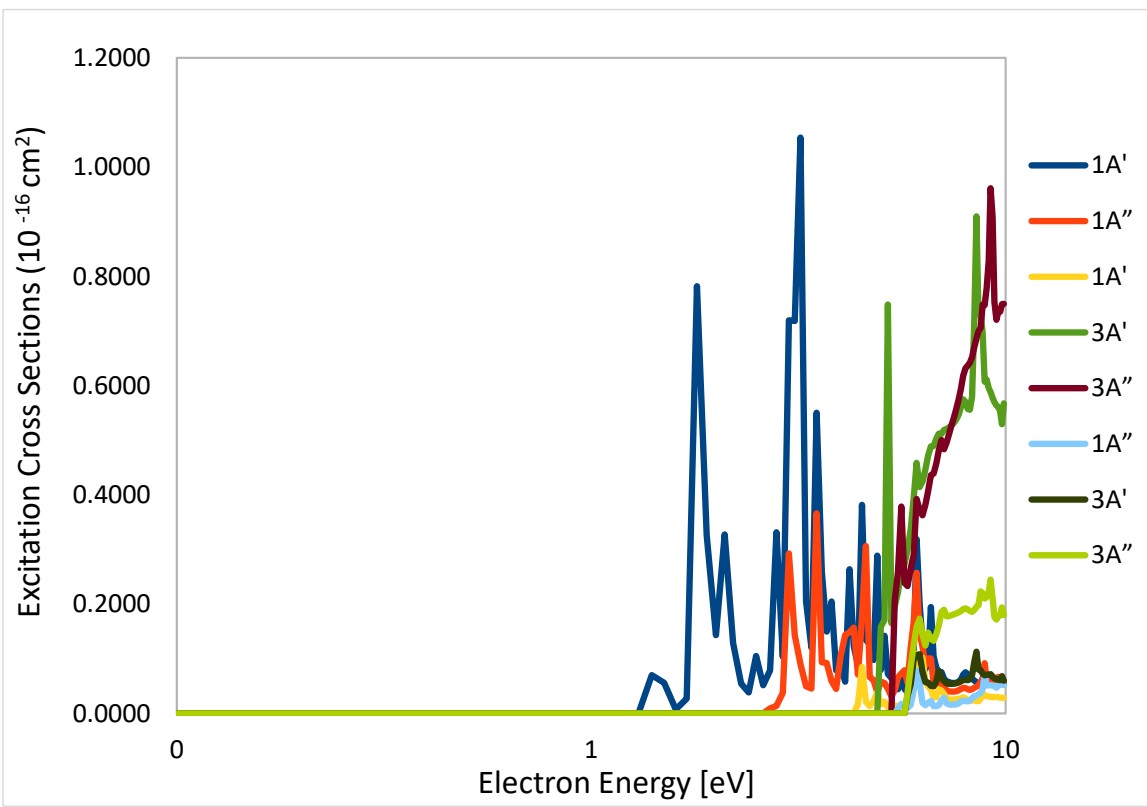

**Figure 3.** Inelastic Scattering Cross Sections for Co(CO)$_3$NO from Quantemol-N simulations.

Co(CO)$_3$NO has 84 electrons and it's ground state is a closed shell $^1$A$_1$ state with C$_{3v}$ symmetry and C$_1$ work point group with its lowest lying excited state $^3$A [17–19]. The Co-NO is at 160.8° and Co-N-O is 4A″ state at 149.5°. With the two bond dissociation energies, for Co-NO bond at a value of 1.63 eV and for the Co-CO bond at 1.26 eV [20], the initial ground state in the fragmentation of Co(CO)$_3$NO is $^3$A″ with the excitation states $^1$A′, $^1$A″, $^1$A′, $^3$A′, $^3$A″, $^1$A″ (Figure 3), $^3$A′, and the $^3$A″ giving the symmetry scattering resonances from the triplet state as $^2$A′, $^2$A″, $^4$A′ and $^4$A″. To simplify our Quantemol-N simulations, a C$_{2h}$ geometry of the molecule was used. The active space from the Quantemol-N simulation is 29A′, 30A′, 13A″, 14A″.

**Table 2.** Negative Ion Formation for Co(CO)$_3$NO, Ni(CO)$_4$ and Cr(CO)$_6$.

| Negative Ion | Incident Electron Energy (eV) [10] | Ion Formation | Incident Electron Energy (eV) [21] | Negative Ion | Incident Electron Energy (eV) [22] |
|---|---|---|---|---|---|
| Co(CO)$_2$NO$^-$ | 0.9 | Ni(CO)$_3^-$ | 0.8 | Cr(CO)$_5^-$ | 0.1 |
| CoCONO$^-$ | 2 | Ni(CO)$_2^-$ | 1.7 | Cr(CO)$_4^-$ | 1.5 |
| CoNO$^-$ | 5 | Ni(CO)$^-$ | 4.6 | Cr(CO)$_3^-$ | 4.7 |
| Co(CO)$^-_3$ | 1.8 | Ni$^-$ | 5.4 | Cr(CO)$_2^-$ | 5.9 |
| Co(CO)$^-_2$ | 3 | | | CrCO$^-$ | 8.5 |
| CoCO$^-$ | 6.4 | | | Cr$^-$ | 8.8 |
| Co$^-$ | 7.1 | | | | |

The ionization energy for the Co(CO)$_3$NO has a value of 8.33 eV for an electron affinity of 0.75 eV, with the ionization threshold at a value of 9.2 eV. The presented reaction energy for the dissociation of the Co(CO)$_3$NO into Co(CO)$_2$NO$^-$ + CO is in the range of 0.65 eV.

The total cross-sections of $Co(CO)_3NO$ (Figure 4) has a maximum of the cross-section at 0.1 eV and a value of $10,560 \times 10^{-16}$ cm$^2$ and decreases asymptotically to 10 eV to a value of $184.2 \times 10^{-16}$ cm$^2$. Convex inflexion points in the cross-section distribution spectrum are observed at a value of the electron energy of 0.2 eV and a value of the cross-section of $5589 \times 10^{-16}$ cm$^2$, and at 0.3 eV and a total cross-section value of $3822.59 \times 10^{-16}$ cm$^2$.

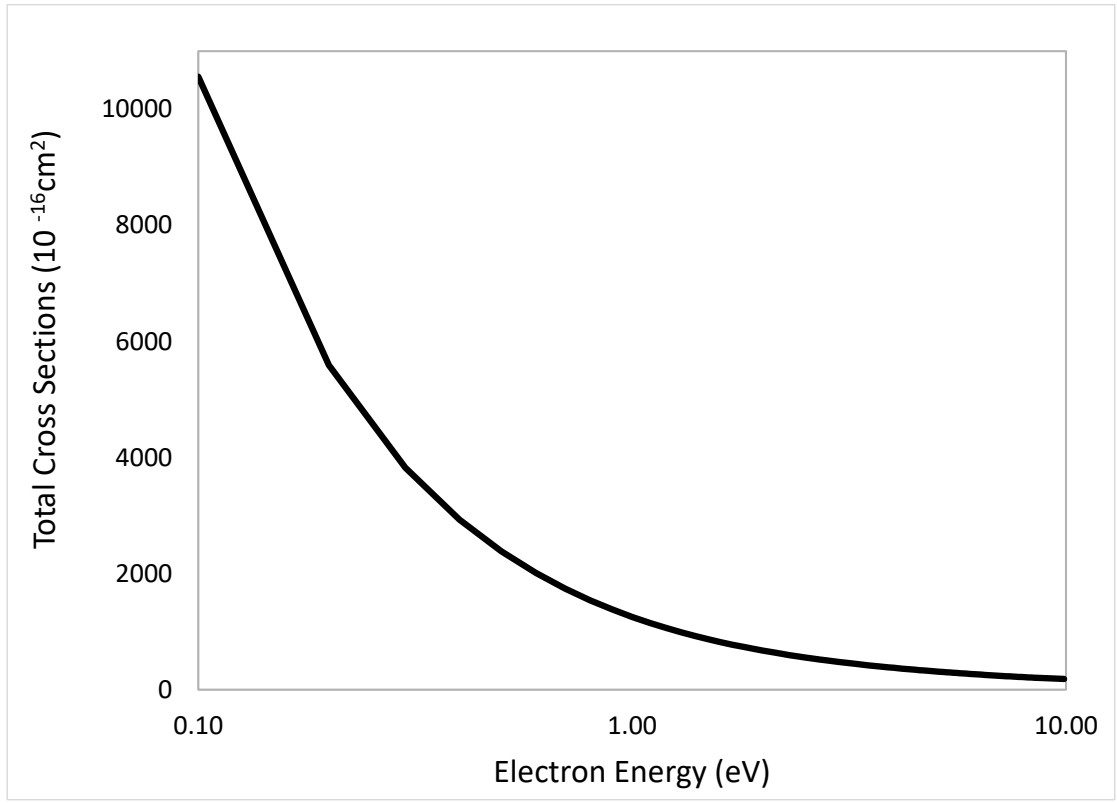

**Figure 4.** $Co(CO)_3NO$ total cross sections from Quantemol-N.

The dissociative electron attachment cross section for $Co(CO)_3NO$ is presented according to the electron affinity, bond dissociation energy and incident electron energy. The electron affinity of the negative ions formed through the dissociative electron attachment process are discussed in [23] with values close to 1.35 eV $<$ EA[$Co(CO)^-_2$] $<$ EA[$Co(CO)(NO)^-$] $<$ EA[$Co(CO)^-_3$] $<$ EA[$Co(CO)_2NO^-$] = 1.73 eV. The simulations were run taking into account these differences.

The maximum of the cross section is in the range of $4.5 \times 10^{-16}$ cm$^2$ at 0.1 eV corresponding to the $Co(CO)_2NO^-$ fragment [24]. The individual peaks in the DEA cross section graph (Figure 5) are corresponding to the individual fragment dissociation, and are close in energy to the values we found in the literature. The maximum cross section value changes with the vibrational frequency. The purple (color) (Figure 3) curve is close to reality, presenting a clean peak at a vibrational frequency of 0.53 Å$^2$ and 1.173 eV electron affinity.

**$Cr(CO)_6$.** The $Cr(CO)_6$ bond distances (Table 3) we used for our simulations are 1.916 Å between Cr-C and 1.171 Å between C-O and the symmetry point group of the ground state molecules is $O_h$, though different sets of values have been reported in [25]. The bond distances reported in $Cr(CO)_6$ are 1.926 Å for in axis Cr-C and 1.139 Å for C-O. The values are reduced for the equatorial bonds for Cr-C with a value of 1.918 Å and C-O with a value of 1.141 Å [25,26]. This values have been used for the structure of the $Cr(CO)_6$ in our DEA cross-sections (Figure 8) and total cross-sections (Figure 6) from Quantemol-N calculations. The symmetry point group of the molecule is $O_h$, but for our simulations simplification we used a $D_{2h}$ configuration.

**Table 3.** Bond distances for Cr(CO)$_6$.

| Molecule | Bond Distances (Å) CCSD (Cr-C) [27] | Bond Distances (Å) CCSD (C-O) [27] | Bond Distances (Å) CCSD (Cr-C)$_{eq}$ [25] | Bond Distances (Å) CCSD (C-O)$_{eq}$ [25] |
|---|---|---|---|---|
| Cr(CO)$_6$ | 3.684 | 2.207 | 1.918 | 1.141 |

The structural and symmetry data used is presented in Appendix A. Whitaker and Jefferey (1967) [28] determined the space group of Cr(CO)$_6$ as the *Pnma* or *Pn2$_1$a*. The dissociation process in Cr(CO)$_6$ follows the steps:

$$Cr(CO)_6 + e^- \rightarrow Cr(CO)_6{}^* \rightarrow Cr(CO)_6{}^- \rightarrow Cr(CO)^-{}_x + n(CO), \text{ where } n = 6 - x \quad (6)$$

The dissociation of Cr(CO)$_6$ into Cr(CO)$_5{}^-$ and a (CO) radical at 0.1 eV, as a result of the dissociative electron attachment (DEA) process, is a transition from the lowest lying LUMO orbital 9a$_{1g}$ ($\sigma$) or one of the higher lying virtual orbitals ($\pi$) to a HOMO higher energy state orbital ($\sigma^*$) or to the highest unoccupied HOMO orbital ($\pi^*$). The optical spectra of Cr(CO)$_6$ presents the only one allowed spin transitions $^1$A$_{1g}$ $\rightarrow$ $^1$T$_{1u}$ as well as multiple other smaller bands assigned to $^1$A$_{1g}$ (2t$^6{}_{2g}$) $\rightarrow$ $^1$T$_{1g}$, $^1$T$_{2g}$ (2t$^5{}_{2g}$ 6e$^1{}_g$) transitions [29,30]. The 3.5 eV to 7 eV was assigned [29] to $^1$A$_{1g}$ $\rightarrow$ $^1$T$_{1u}$, while the 4.83 eV was assigned to $^1$T$_{2g}$ and 4.91 eV to $^1$A$_{1g}$ $\rightarrow$ a$^1$T$_{2g}$, 3.60 eV and 3.91 eV was assigned to the allowed transition $^1$A$_{1g}$ $\rightarrow$ a$^3$T$_{1u}$.

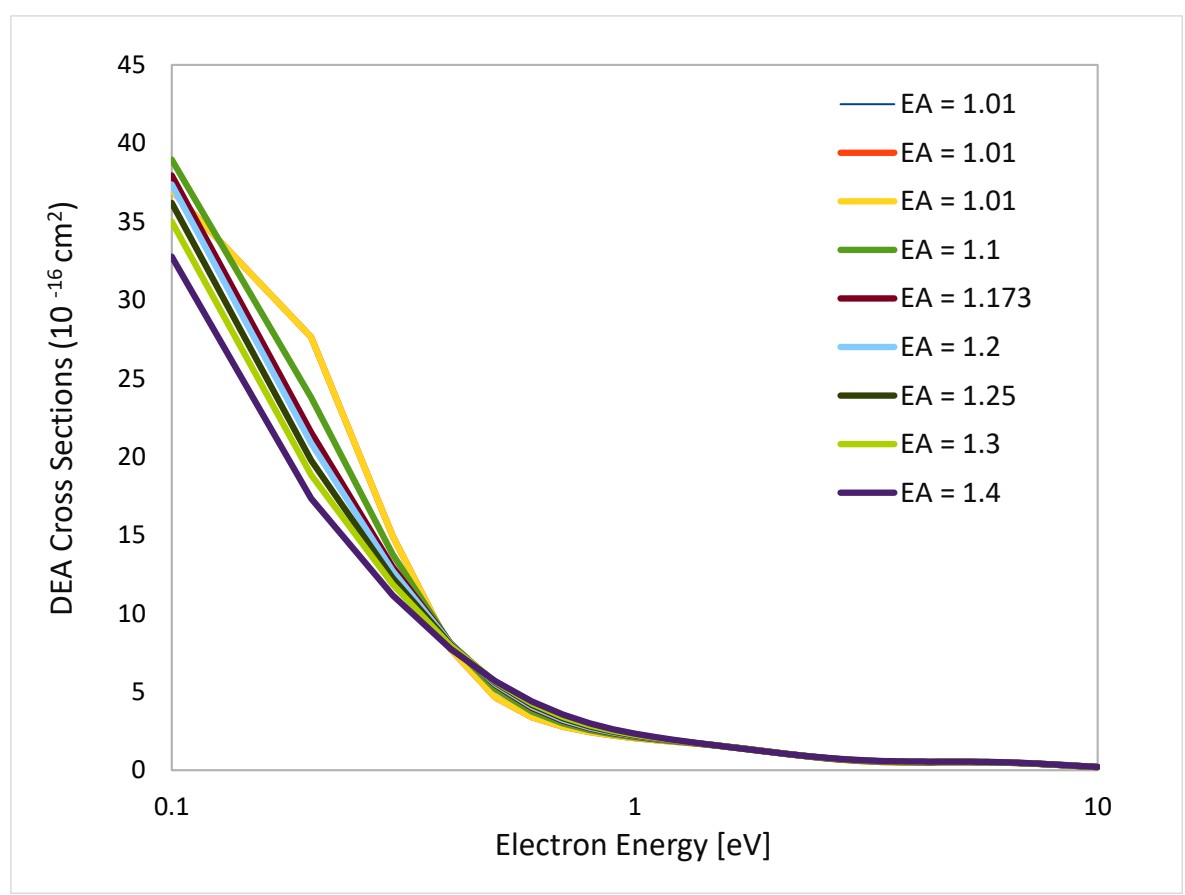

**Figure 5.** Dissociative electron attachment cross sections from Co(CO)$_3$NO.

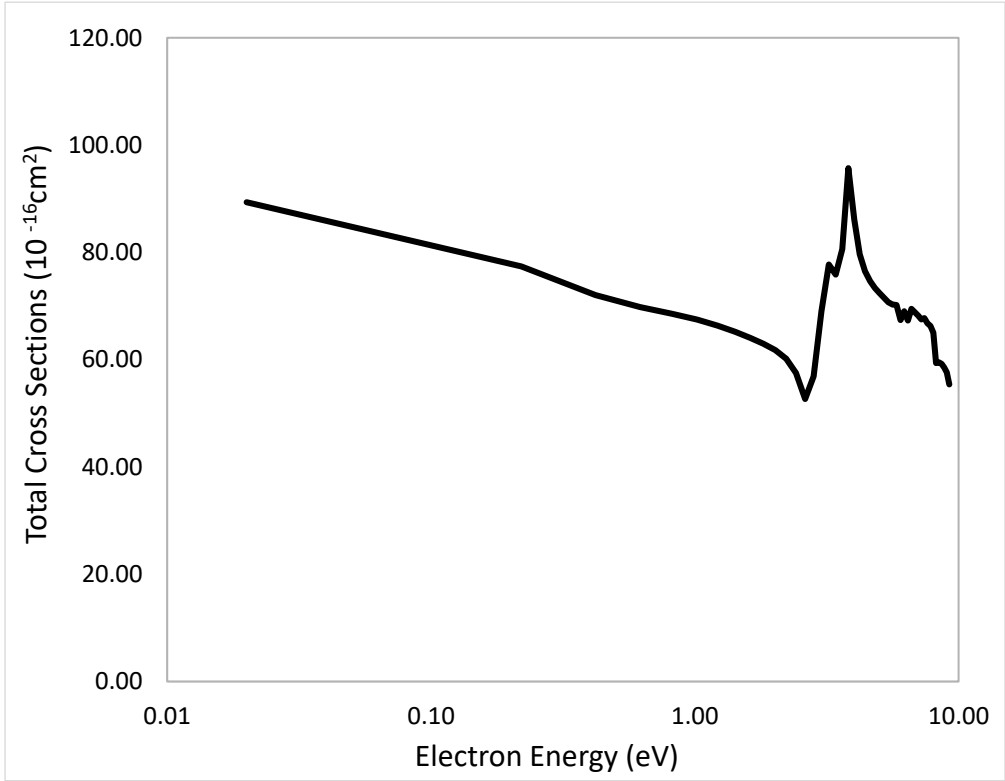

**Figure 6.** $Cr(CO)_6$ total cross sections from Quantemol-N.

Villaume et al. (2007) [31] defines the structure of the molecule as $D_{2h}$ and the electronic ground state $^1A_{1g}$ as having the electronic configuration $(8a_{1g})^2(7t_{1u})^6(1t_{2g})^6(1t_{2u})^6(1t_{1g})^6(5e_g)^4$ $(8t_{1u})^6(2t_{2g})^6$. At 4.48 eV and 4.50 eV, the allowed transition is a $a^1A_{1g} \rightarrow {}^1T_{2u}$. At a temperature of 77 K, vibrational bands appear at 4.05 eV, 4.49 eV; at 300 K the vibrational bands can be found at 5.53 eV, 4.87 eV and 6.36 eV, all assigned to $a^1A_{1g} \rightarrow {}^1T_{1g}$ and $a^1A_{1g} \rightarrow {}^1T_{2g}$.

The ionization potential for the ions of the $Cr(CO)_6$ molecule have values higher than the threshold value of ~8 eV. Table 4 presents the ionization potential of the parent $Cr(CO)_6$ molecule from the publications of Winters and Kiser (1965) [32], Fukuda et at (2009) [33], Foffani et al. (1964) [34] showing a ionization potential for carbonyls with a value of 1.5 eV higher than for the metal atom and similar values within 0.1–0.5 eV. The basis set used for our Quantemol-N calculations on cross sections are cc-pVDZ for C atom, 6–311G for O atom and cc-pVTZ on Cr.

**Table 4.** Ionization potential of $Cr(CO)_6$.

| Compound | Fukuda et al. (2009) [33] | Winters and Kiser (1965) [32] | Foffani et al. (1964) [34] | Electron Ionization [35] | Photon Impact [35] | Metal Atom [35] | Junk and Svec (1968) [35] |
|---|---|---|---|---|---|---|---|
| $Cr(CO)_6$ | 8.5 | $8.15 \pm 0.17$ | $8.18 \pm 0.07$ | 8.23 | 8.03 | 6.76 | $8.44 \pm 0.05$ |

The DFT calculations in [36] reveal the vibrational excitation bands due to the photoionization of the molecule and show the process as being a transition from $de_g^*$ orbitals in σ-antibonding with the metal and a reduction in the number of electrons in π-bonding $t_{2g}$ orbitals to occupying the $Cr\text{-}dz^2\text{-}CO\text{-}5\sigma$ $e_g^*$orbitals. The excitation cross sections data (Figure 7) was calculated using Quantemol-N simulations using dissociation electron energy data and electron affinity of the formed negative ions presented in Table 5.

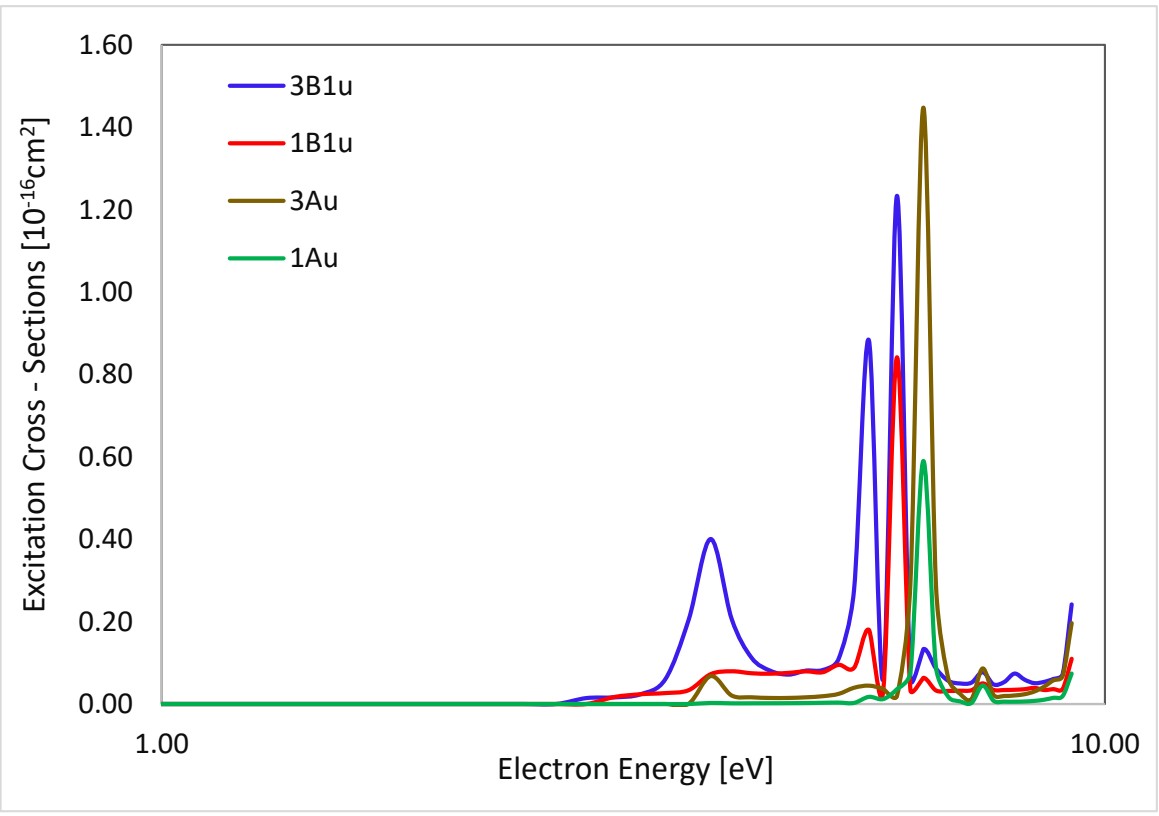

**Figure 7.** Excitation cross sections of $Cr(CO)_6$.

**Table 5.** Negative ions of $Cr(CO)_6$ with electron affinity.

| Negative Ion | Dissociation Mechanism (eV) from $Cr(CO)_6$ | Experimental Incident Electron Energy (eV) [22] | Electron Affinity (eV) | Vibrational Frequency (cm$^{-1}$) |
|---|---|---|---|---|
| $Cr(CO)_5{}^-$ | $Cr(CO)^-{}_5 + CO$ | 0.1 | >1.6 eV [34] | 2000 |
| $Cr(CO)_4{}^-$ | $Cr(CO)^-{}_4 + 2(CO)$ | 1.5 | | |
| $Cr(CO)_3{}^-$ | $Cr(CO)^-{}_3 + 3(CO)$ | 4.7 | | |
| $Cr(CO)_2{}^-$ | $Cr(CO)^-{}_2 + 4(CO)$ | 5.9 | | |
| $CrCO^-$ | $Cr(CO)^- + 5(CO)$ | 8.5 | | |
| $Cr^-$ | $Cr^- + 6(CO)$ | 8.8 | | |

If the $Cr(CO)_6$ molecule is seen in $C_{2v}$ point group [25,37,38], the allowed transition from $1^1A_1$ to $T_{2u}$ state is characterized by 3 sub-states: $1^1B_1$, $1^1B_2$ and $2^1A_1$, where $1^1B_1$ and $1^1B_2$ are degenerate states corresponding to $1^1E$ in a $C_{4v}$ point group. A Cr-C bond distance higher than 0.25 Å the energy levels of the transition states are: $E(1^1B_1) = E(1^1B_2) > E(1^1A_2)$ for $C_{2v}$ point group and $E(1^1E) < E(1^1B_2)$ for $C_{4v}$ point group. The vibrational and transition states data is presented in Table 5.

The DEA cross sections are presented in Figure 8, simulated with different electron affinity for comparison in the spectrum changes. The values used for the electron affinity of the negative ions of $Cr(CO)_6$ are expected to be higher than the specified value in [39] of 1.6 eV. The DEA cross section maximum value is in the range of ~$4.3 \times 10^{-15}$ cm$^2$ at 0.2 eV for the $Cr(CO)_5{}^-$ fragment, for a value of electron affinity of 1.6 eV and vibrational frequency of 2000 cm$^{-1}$ for a dissociation electron energy value of 0.1 eV. The value of cross-sections reported by [25] is of 1.85 Å$^2$ to 3.29 Å$^2$. The ground state configuration used in our Quantemol-N calculations in $^1A_1$ state is $D_{2h}$. The final assigned states for the DEA

process as a result of the calculations from the initial $^1A_g$ state are: $^3B_{1u}$, $^1B_{1u}$, $^3A_u$, $^1A_u$ with 5 symmetry states (Table 6). From Quantemol-N simulations the resonances from our active space are: $^2A_g$ (4.86 eV), $^2B_{3u}$ (9.04 eV), $^2B_{2u}$ (9.04 eV), $^2B_{1g}$ (8.1 eV), $^2B_{1u}$ (8.1 eV), $^2B_{2g}$ (8.3 eV), $^2B_{3g}$ (8.3 eV) and $^2A_u$ (6.7 eV).

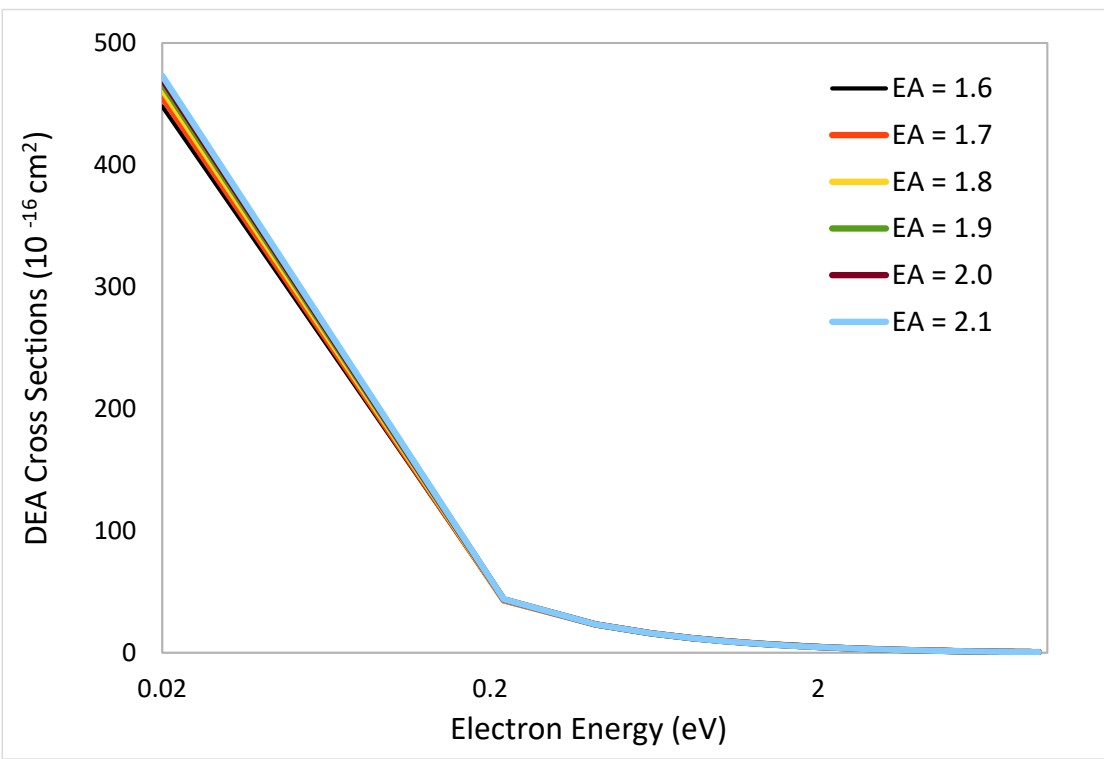

**Figure 8.** DEA cross sections of $Cr(CO)_6$.

**Table 6.** Vibrational excitation energies of $Cr(CO)_6$; (Q-N* used for Quantemol-N simulations).

| State | TDDFT/B3LYP–ΔE (eV) [25] | Experimental [25] ΔE (eV) | Experimental [22]/Our Q-N* Sim. ΔE (eV) |
|---|---|---|---|
| $1^1E_u$ | 4.14 | | |
| $1^1T_{2u}$ | 4.2 | | |
| $1^1A_{2u}$ | 4.25 | | |
| $1^1T_{1u}$ | 4.5 | 4.44 | 4.7 |
| $1^1T_{1g}$ | 4.65 | | |
| $1^1A_{1u}$ | 4.7 | | |
| $2^1T_{2u}$ | 4.82 | | |
| $2^1E_u$ | 4.71 | | |
| $1^1E_g$ | 4.99 | | |
| $1^1T_{2g}$ | 4.74 | | |
| $2^1T_{1g}$ | 4.91 | | |
| $2^1T_{2g}$ | 5.59 | | |
| $2^1T_{1u}$ | 6.02 | 5.48 | 5.9 |

**Ni(CO)₄.** The $Ni(CO)_4$ [39–42] is a tetrahedral, $D_h$, with its ground state $^1A_1$. The Ni - n(CO) bond is a d → π* transition, from the lower stable orbitals d orbitals with ground state ($^1A_1$) to the higher π* orbitals of σ-symmetry with the ($1^1T_1$, $1^1T_1$, $3^1T_1$, $1^1E$, $2^1T_2$)

excited states [27]. The UV spectrum of Ni(CO)$_4$ in [43] has peaks at 6.0 eV, 5.4 eV and 4.6 eV, representing transition dominated by d $\rightarrow$ 2$\pi$* for 6.0 eV, $^1$A$_1$ $\rightarrow$ $^5$T$_2$ for 5.36 eV close to the 5.4 eV value, representing a d $\rightarrow$ $\pi$* transition. From 3.36 eV to 3.94 eV, the transition spectrum is dominated by d $\rightarrow$ $\pi$*, (A$_1$, E, T$_1$, T$_2$) to (T$_1$, T$_2$). The absorption cross-sections value for Ni(CO)$_4$ is in the range of ~5.01 $\times$ 10$^{-17}$ cm$^2$. Excitation cross sections derived from our Quantamol-N calculations are presented in Figure 9.

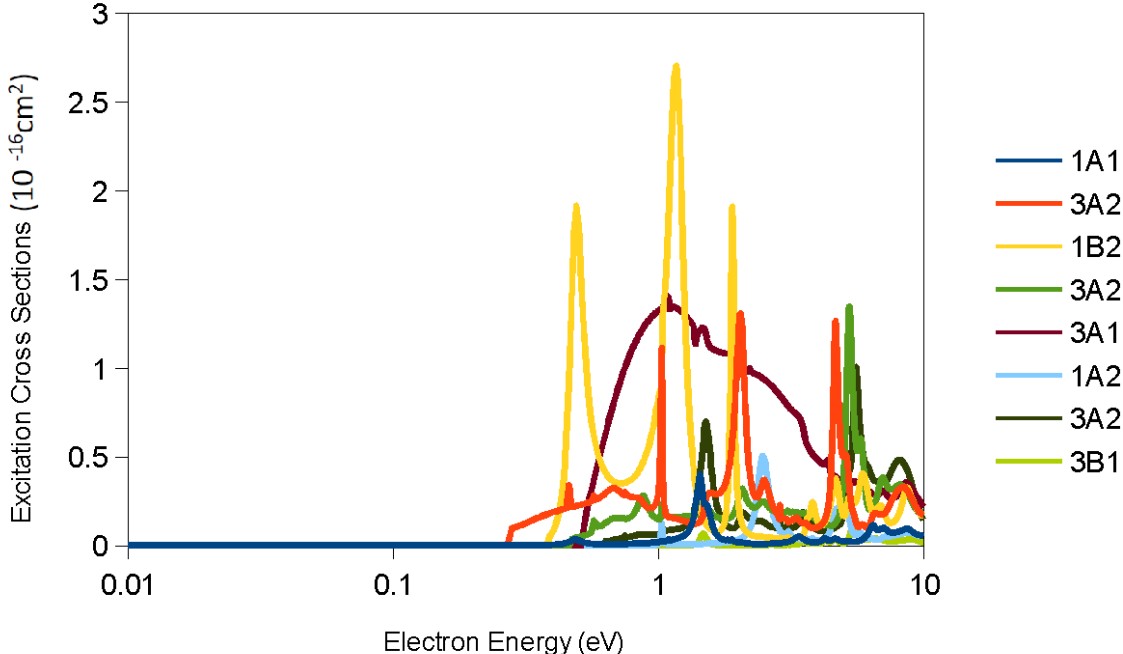

**Figure 9.** Excitation cross sections of Ni(CO)$_4$.

For Ni(CO)$_4$, the dissociation process follows the reaction equation:

$$\text{Ni(CO)}_4 + e^- \rightarrow \text{Ni(CO)}_4^* \rightarrow \text{Ni(CO)}^-_x + n\text{(CO)}, n = 4 - x \qquad (7)$$

DEa to Ni(CO)$_4$ is known to strip all four ligands, so we expect to see resonances specific to the molecule undergoing full dissociation, with the negative ions, Ni(CO)$_3$$^-$, Ni(CO)$_2$$^-$, NiCO$^-$ and Ni$^-$, at energies between 0.5–6 eV. As one electron collides with one molecule in ground state d and excites it to a higher excited state $\pi$* followed by fragmentation, undergoing a allowed transition, the kinetic energy of the molecule increases from E$_G$ to E$_G$$'$ to drop further to E$_{KER}$$'$ and E$_{KER}$$''$ > E$_G$, as we take the fragmentation process as a step by step process. The active space configuration of the theoretical model from our Quantemol-N calculations is 10A$_g$, 11A$_g$, 12A$_g$, 10A$_u$, 11A$_u$, 10B$_u$, 11B$_u$, 12B$_u$, 11B$_g$.

The excited states obtained as a result of the collisions between the electron and the Ni(CO)$_4$ molecule representing the active space are 1A$_1$, 3A$_2$, 1B$_2$, 3A$_2$, 3A$_1$, 1A$_2$, 3A$_2$ and 3B$_1$. The excited states have the appearance energies over 0.1 eV with the first excited state present at an electron energy of 0.6 eV (3A$_2$) involved in the fragmentation of the Ni(CO)$_4$ with the formation of the highest mass anion Ni(CO)$_3$$^-$. Other excited states involved in the fragmentation of the compound around 0.8 eV energy are 1B$_2$ (0.7 eV), 3A$_2$ and 1A$_1$ (0.75 eV), 3A$_2$ (0.8 eV) and 3A$_1$ (0.85 eV). Higher excited states fall around 1 eV electron energy and are involved in the fragmentation of the Ni(CO)$_4$ with the formation of lower mass anions Ni(CO)$_2$$^-$, NiCO$^-$ and Ni$^-$.

The molecular structure with symmetry in Cartesian coordinates (X, Y, Z) used for the cross section simulation parameters is presented in Appendix A. The bond distances between Ni-C and C-O have been reviewed from multiple sources, in [44] having the values of 1.669 Å for Ni-C and 1.153 Å for C-O. The bond distances determined in [3]

experimentally and through calculations have the value of 1.838 Å for Ni-C and 1.142 Å for C-O from experiment [45] and 1.831 Å for Ni-C and 1.147 Å for C-O determined by CCSD calculations using cc-pVTZ basis set.

In [46] the Ni-C and C-O bond lengths have a value of 1.84 Å and 1.15 Å from X-ray single crystal study, and 1.835 Å and, respectively 1.139 Å from gas-phase electron diffraction. A $C_{2h}$ geometry was employed to reduce the calculations steps and the number of iterations. Our Quantemol-N simulations are using a user defined basis set, based on cc-pVTZ and STO-6G. The basis set was user defined to reduce the size of the data and the memory necessary for the simulations. The values of electron affinity of $Ni(CO)_4$ used for simulation are presented in Table 7.

**Table 7.** Negative ions of $Ni(CO)_4$ with electron affinity [40].

| Negative Ion | Incident Electron Energy (eV) [21] | Appearance Potentials (eV) [47] | Electron Affinity (eV) [47] | Vibrational Frequency (cm$^{-1}$) [47] |
|---|---|---|---|---|
| $Ni(CO)_3^-$ | 0.8 | 0 | $0.804 \pm 0.012$ | $2100 \pm 80$ |
| $Ni(CO)_2^-$ | 1.7 | $1.0 \pm 0.4$ | $0.643 \pm 0.014$ | $2100 \pm 80$ |
| $Ni(CO)^-$ | 4.6 | $3.2 \pm 0.5$ | $1.077 \pm 0.013$ | $1940 \pm 80$ |
| $Ni^-$ | 5.4 | $4.1 \pm 0.3$ | $1.157 \pm 0.010$ | |

The DEA cross section (Figure 10) from Quantemol-N simulations, presented in Figure 5, maximum cross section of $1-2 \times 10^{-3}$ Å$^2$ at 0.8 eV corresponding to the $Ni(CO)_3^-$ fragment, representing the curves with electron affinity values between 1.2 to 1.4 eV. Comparatively, taking into account the fragmentation pattern, the violet (color) spectrum (Figure 10) has better accuracy and it is closer to the true value, though all present reliable data within the error limit threshold according to the EA(eV) and vibrational frequency used.

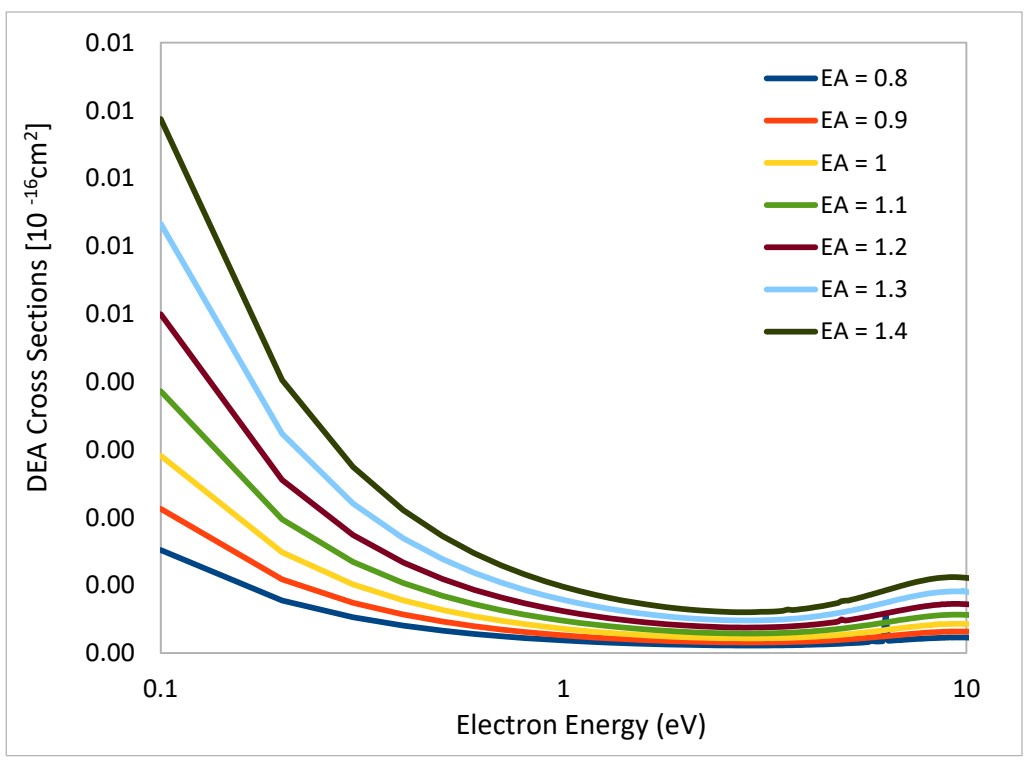

**Figure 10.** DEA cross sections of $Ni(CO)_4$.

With the increase in the electron affinity we have an increase in the cross-section values as seen from Figure 6.

The total cross-section (Figure 11) from Quantemol-N simulations is in close agreement with the reported literature data for ~100–300 eV of $2 \times 10^{-16}$ cm$^2$ [48,49], with a value of $1.6$–$1.8 \times 10^{-16}$ cm$^2$ in the same energy range. The experimental work in [49] makes reference to the same cross-section value of $2 \times 10^{-16}$ cm$^2$ in the energy range of 100–300 eV.

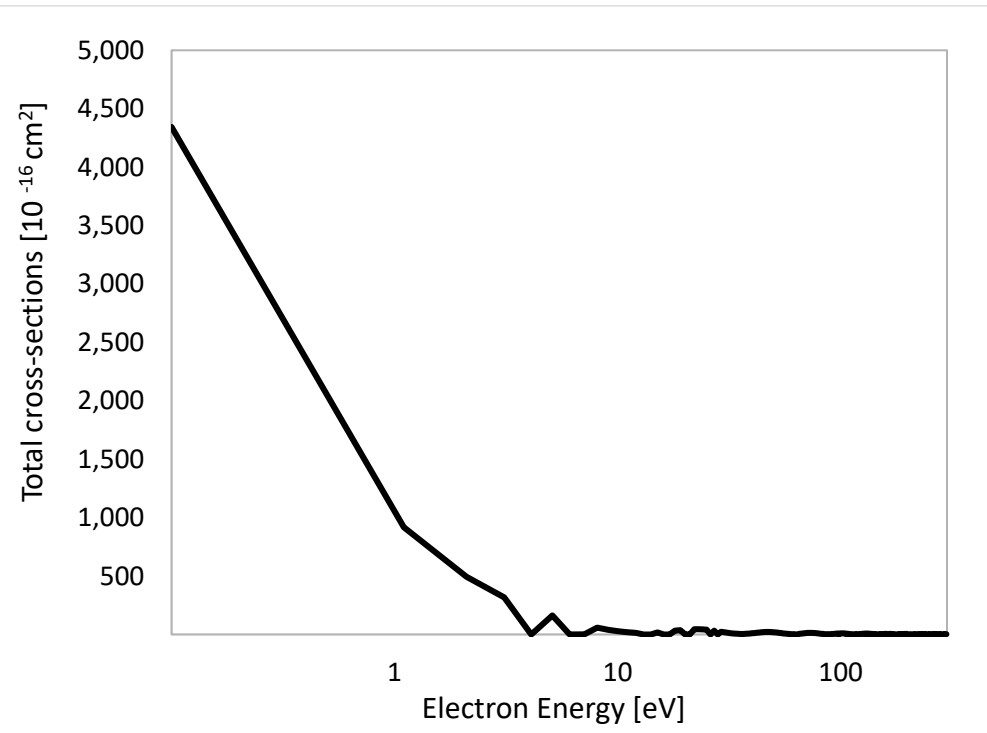

**Figure 11.** Total cross-sections values for Ni(CO)$_4$, 1 eV to 300 eV.

## 5. Conclusions

With increasing importance of metal containing organic compounds in cancer research, materials and superconductors development, nanotechnology and focused electron beam deposition there is a need to provide electron impact cross section data for such compounds and in particular values of dissociative electron attachment. In thisd paper we present electron scattering cross sections derived usnf Quantemol-N for three of the most commonly used commercial materials: Ni(CO)$_4$, Co(CO)$_3$NO and Cr(CO)$_6$.

As the use of Quantemol-N simulations would not necessarily replace experimental data, the results show promising use of the software for reliable cross section data that can be utilized in the multitude of the industrial processes requiring an accurate value of these. The approach we employed, based on R-matrix calculations is rather a simple method making use of basic molecular data, structure and symmetry of the molecule, bonds lengths and specificity, all available and easy to obtain from literature or experimental data.

**Author Contributions:** The authors contributed in equal parts to the article. All authors have read and agreed to the published version of the manuscript.

**Funding:** We want to acknowledge MP and NJM receiving funding from the European Union's Horizon 2020 research and innovation program under the Marie Skłodowska-Curie grant agreement No 722149.

**Data Availability Statement:** All data will be provided upon reasonable request.

**Conflicts of Interest:** The authors declare no conflict of interest.

## Appendix A

**Table A1.** Atomic structure and X, Y, Z configuration of Cr(CO)$_6$.

| Atom Label | X [Å] | Y [Å] | Z [Å] |
| --- | --- | --- | --- |
| C1 | 1.34 | 1.34 | 0 |
| C2 | −1.34 | −1.34 | 0 |
| C3 | 0 | 0 | 1.9 |
| C4 | −1.34 | 1.34 | 0 |
| C5 | 0 | 0 | 0 |
| C6 | 2.15 | 2.15 | 0 |
| Cr7 | −2.15 | −2.15 | 0 |
| O8 | 0 | 0 | 3.04 |
| O9 | −2.15 | 2.15 | 0 |
| O10 | 0 | 0 | −3.04 |
| O11 | 2.15 | −2.15 | 0 |
| O12 | 0 | 0 | −3.04 |
| O13 | 0 | −3.04 | 0 |

**Table A2.** Atomic structure and X, Y, Z configuration of Co(CO)$_3$NO.

| Atom Label | X [Å] | Y [Å] | Z [Å] |
| --- | --- | --- | --- |
| Co1 | −0.1 | 0 | 0 |
| C2 | 0.66 | −0.81 | −1.4 |
| C3 | 0.66 | −0.81 | 1.4 |
| C4 | 0.66 | 1.62 | 0 |
| O5 | 1.12 | −1.34 | 2.32 |
| O6 | 1.12 | −1.34 | −2.32 |
| O7 | 1.11 | 2.68 | 0 |
| O8 | −2.92 | 0 | 0 |
| N9 | −1.76 | 0 | 0 |

**Table A3.** Atomic structure and X, Y, Z configuration of Ni(CO)$_4$.

| Atom Label | X [Å] | Y [Å] | Z [Å] |
| --- | --- | --- | --- |
| Ni1 | 0 | 0 | 0 |
| C2 | −0.09 | −1.8 | 0.18 |
| C3 | 1.73 | 0.52 | 0.04 |
| C4 | −0.74 | 0.48 | −1.58 |
| C5 | −0.9 | 0.79 | 1.35 |
| O6 | −0.15 | −2.94 | 0.3 |
| O7 | 2.83 | 0.85 | 0.07 |
| O8 | −1.2 | 0.79 | −2.58 |
| O9 | −1.47 | 1.29 | 2.21 |

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
