# Peer review of "Dissociative Electron Attachment Cross Sections for Ni(CO)4, Co(CO)3NO, Cr(CO)6"

_chemistry, doi:10.3390/chemistry4030072_

Round 1

Reviewer 1 Report

There are many typos, and the article is written in a bit careless way.

A few points to correct / clarify:

Page 1, line 15/16 - at what energy? Is it a maximum signal between 0-20 eV?

Page 2, line 41/42 - three?

Page 2, line 45-48 - two separate sentences?

Page 2, line 59

Page 2, line 62

Page 3, line 79

Page 3, line 107/108 - the maximum?

In Fig. 3 and 6, 7, 9 it would be nice to see the energy scale 0-20 eV, not up to 10 eV

Not really know what is the point presenting Fig. 11 as it is.

I assume all atomic/molecular structures  were optimized. What basis sets and methods structures X,Y,Z are based on?

Author Response

I updated all the minor changes to the article.

Regarding the energy scale of the graphs from 0 to 20 eV, we cannot update the scale as the files were recorded from 0 to 10 eV. We decided to record them at 0 to 10 eV as no anions are declared at energies higher than 9 eV. We had to take that decision as the computations are resource-demanding and there was no data to support the decision of running such files. 

I introduced a new section to the article named "Computational Details" where I added a few details about the parameters used for the cross-sections.

Author Response

I updated the minor changes.

1. I looked into ways of making the graphical abstract more informative.

2. I updated FEBID.

3. I fixed the double "scattering".

4. I added the AB- anion to the equations.

5. I updated the phrase in line 76.

6. I fixed the double "is".

7. I fixed 1.9 and 1.10, now 1.8 and 1.9.

Round 2

Reviewer 2 Report

The manuscript may be published in its present form. However, the authors should clearly distinguish between Graph abstract and Figure 1. The latter is still not mentioned in the text.